# Numerical Simulation of Effect of Different Initial Morphologies on Melt Hydrodynamics in Laser Polishing of Ti6Al4V

**DOI:** 10.3390/mi12050581

**Published:** 2021-05-20

**Authors:** Kai Li, Zhenyu Zhao, Houming Zhou, Hao Zhou, Jie Yin, Wei Zhang, Guiyao Zhou

**Affiliations:** 1School of Intelligent Manufacturing and Equipment, Shenzhen Institute of Information Technology, Shenzhen 518172, China; 201821541904@smail.xtu.edu.cn (K.L.); 201821541865@smail.xtu.edu.cn (H.Z.); 201921001888@smail.xtu.edu.cn (J.Y.); zhang@yeah.net (W.Z.); 2School of Mechanical Engineering, Xiangtan University, Xiangtan 411105, China; 3Guangdong Provincial Key Laboratory of Nanophotonic Functional Materials and Devices, South China Normal University, Guangzhou 510006, China; Zguiyao@163.com

**Keywords:** laser polishing, different surface morphologies, melt hydrodynamics, numerical simulation

## Abstract

As a surface finishing technique for rapid remelting and re-solidification, laser polishing can effectively eliminate the asperities so as to approach the feature size. Nevertheless, the polished surface quality is significantly sensitive to the processing parameters, especially with respect to melt hydrodynamics. In this paper, a transient two-dimensional model was developed to demonstrate the molten flow behavior for different surface morphologies of the Ti6Al4V alloy. It is illustrated that the complex evolution of the melt hydrodynamics involving heat conduction, thermal convection, thermal radiation, melting and solidification during laser polishing. Results show that the uniformity of the distribution of surface peaks and valleys can improve the molten flow stability and obtain better smoothing effect. The high cooling rate of the molten pool resulting in a shortening of the molten lifetime, which prevents the peaks from being removed by capillary and thermocapillary forces. It is revealed that the mechanism of secondary roughness formation on polished surface. Moreover, the double spiral nest Marangoni convection extrudes the molten to the outsides. It results in the formation of expansion and depression, corresponding to nearby the starting position and at the edges of the polished surface. It is further found that the difference between the simulation and experimental depression depths is only about 2 μm. Correspondingly, the errors are approximately 8.3%, 14.3% and 13.3%, corresponding to Models 1, 2 and 3, respectively. The aforementioned results illustrated that the predicted surface profiles agree reasonably well with the experimentally measured surface height data.

## 1. Introduction

Ti6Al4V alloy has been widely used in aerospace [1], chemical industries [2] and medical sector [3] due to its low density [4], high strength [5] and excellent biocompatibility [6]. Therefore, the surface quality, such as surface roughness, of the devices fabricated from Ti64 is posed in a great challenge. At present, the polishing of Ti64 surface is mainly by traditional methods such as mechanical polishing, electrochemical polishing and plasma polishing. It is difficult to polish the complex surface. Laser polishing (LP) as a non-contact processing surface treatment technology, with high polishing efficiency, non-pollution and a high degree of processing flexibility. Additionally, LP not only can effectively surface roughness, but also the polished surface can be secondary strengthening such as hardness [7,8], wear and corrosion resistance [9,10]. Additionally, polishing on Ti6Al4V is a real challenge, which can modify the surface and becoming fragile and Widmanstätten patterns during PAM process [11,12]. This work can prevent the realization of such damage or wrong patterns.

Nonetheless, LP is a complex physical process. Owing to the extremely short interaction duration of the laser with the material, it can reach the millisecond or even microsecond scale. Thereby, it is difficult to observe the liquid metal flow and the evolution of the polished surface topography. Hence, a large number of scholars have predicted surface morphology and melt hydrodynamics by means of FEM models. The concept of surface micro-melting was first introduced by Mai et al. [13]. A hybrid numerical model was developed based on the advantages of variable and fixed domains to calculate the nonlinear problem of metal solid-liquid moving boundary. Based on the damped oscillatory behavior generated by the molten surface, Perry et al. first introduced the one-dimensional critical frequency *f_cr_* to predict the polishing effect of the polished surface within the spatial frequency domains [14,15,16]. When the spatial frequency amplitude of the polished surface is greater than the *f_cr_*, the surface roughness is significantly decreased. Vadali et al. extended the concept of one-dimensional critical frequency to the two-dimensional plane based on a series of physical equations that can predict the spatial frequency content and surface roughness after polishing [17]. The *f_cr_* was accurately obtained by solving the heat conduction differential equation by Ukar et al. [18]. It also predicted the polished surface morphology well. Further, Wang et al. developed a surface prediction model for thermocapillary regime smoothing [19]. They applied the capillary force prediction model proposed by Vadali et al. [17] to predict the spatial spectrum of polished surface by using introduced feature slope and normalized average displacement.

The above simulation process is only a semi-empirical surface prediction model that cannot simulate the evolution of the melt hydrodynamics. Sim et al. developed a numerical model of the axisymmetric thermocapillary flow during laser melting and further analyzed the evolution of the molten free surface with radiation duration [20]. Nevertheless, the solidified surface morphology cannot be determined due to the cooling process is not considered. Ma et al. proposed a two-dimensional (2D) axisymmetric transient model dominated by Marangoni convection. Since the model assumed that the polished surface is an ideal smooth surface [21]. Thus, only Marangoni flow was contained in the molten pool and no capillary flow associated with surface curvature was introduced. In further, Zhang et al. performed a model illustrates the molten flow behavior and the formation of free surface during LP and the contribution of capillary and thermocapillary forces to the melt hydrodynamics were analyzed [22]. The result indicates that capillary forces dominate in the initial stage of melting, mainly eliminating the surface with large curvature. Correspondingly, the thermocapillary force is dominant in the melt development stage and further improves the quality of polished surface. However, there are numerous factors affecting the polishing process. The influence of different initial surface morphologies on the evolution of melt hydrodynamics has been few investigated.

In the present work, a transient 2D model based on a moving heat source is developed by coupling the heat transfer field and the flow field to simulate the evolution of the liquid metal surface from rough to smooth. The mapping law of the melt hydrodynamics with different initial surface features was detailed examined from three aspects: temperature field, velocity field and the evolution of different molten surface profiles. In particular, the underlying cause of the bumps formation during polishing process was analyzed. To verify the numerical model, laser polishing experiments were performed on three different surfaces as well as polished surface were measured for comparison with the predicted surface profile.

## 2. Numerical Simulation

### 2.1. Governing Equations

To ensure the accuracy while saving computational costs, the laser polishing model was developed based on the following assumptions.

(a)The property of fluid phase fluid is treated as incompressible Newtonian laminar flow.(b)The material distribution satisfies continuity and isotropy.(c)The laser incident energy is considered as the surface heat flux.(d)Due to the ratio of the density of the liquid Ti6Al4V and the dynamic viscosity argon gas of is large, the influence of gas flow on the free surface evolution can be neglected.

In this paper, the model is based on the energy equation (Equation (1)), momentum equation (Equation (2)) and continuity equation (Equation (3)) as the theoretical guidance [22,23,24,25,26].
(1)$$\rho C_{p}^{*}[\frac{\partial T}{\partial t}+\nabla \cdot (\vec{u}T)]-\nabla \cdot (k\nabla T)=0$$
(2)$$\rho \left(\frac{\partial \vec{u}}{\partial t}+\left(\vec{u}-{\vec{u}}_{m}\right)\cdot \left(\nabla \vec{u}\right)\right)=\nabla \cdot \left(-pI+\mu \left(\nabla \vec{u}+{\left(\nabla \vec{u}\right)}^{T}\right)\right)+F_{V}$$
(3)$$\nabla \cdot \vec{u}=0$$
where *ρ* is the density, *t* is the laser radiation duration, *T* is the the variation of surface temperature with *t*, $\vec{u}$ is the melt velocity, *k* is the thermal conductivity, ${\vec{u}}_{m}$ is the mesh velocity, *p* is the pressure, *I* is the identity matrix, *μ* is the dynamic viscosity, *F_V_* is the body force of buoyancy and gravity of the molten pool [23].
(4)$$F_{V}=\rho_{ref}g_{}\left(1-\beta \left(T-T_{ref}\right)\right)$$
where *T_ref_* is the reference temperature, *ρ_ref_* is the reference density, *β* is the thermal expansion coefficient and *g* is the gravity constant.

To balance the energy, the latent heat of melting term released during the solid to liquid phase transition. It is added to the specific heat capacity function as an equivalent heat capacity using a Gaussian function in the simulation [22].
(5)$$C_{p}^{*}=C_{p}+L_{m}(\frac{df_{L}}{dt})$$
where *C_p_* is the specific heat, *L_m_* is the latent heat of melting, the definition of liquid fraction *f_L_* as follow [22,23,24,25,26]
(6)$$f_{L}=\{\begin{array}{c} \begin{array}{cc} 0 & T\leq T_{s} \end{array}\\ \begin{array}{cc} \frac{T-T_{s}}{T_{l}-T_{s}} & T_{s}\leq T\leq T_{l} \end{array}\\ \begin{array}{cc} 1 & T_{l}\leq T \end{array} \end{array}$$
where *T_s_* is the solid phase temperature, *T_l_* is the liquid phase temperature.

In addition, Table 1 illustrates the thermophysical properties of Ti6Al4V are used during the modeling process.

### 2.2. Model Geometry

Ramos et al. assumed that the free surface of selective laser sintering part is composed of closed hemispherical caps during study SSM mechanism of LP [28,29,30,31]. However, the experimental polished surface is more asperities, consisting of various irregular peaks and valleys. In order to establish an accurate geometric model, firstly, a 3D optical profilometer based on white light interferometer technology was used for initial surface profile inspection and results as shown in Figure 1a. Then, due to high frequency noise of 3D optical surface profile will cause the odd occurrence of surface during the geometric modeling process. Thus, a rectangular window function is added to satisfy the Fourier filtering of the surface high frequency features, so as to obtain smooth geometric features closer to the experimental surface. The filtered surface morphology is shown in Figure 1b.

Furthermore, to investigate the effect of different initial surface morphologies on the molten pool characteristics, three different linear surface profiles were extracted from the filtered surface, as shown in Figure 1c,e,g. The black and red lines are, respectively, representing the initial and the filtered linear profile. Then, the filtered surface data is imported into COMSOL by means of the interpolation function to build the geometric model, which is defined as Model 1 (see Figure 1d), Model 2 (see Figure 1f) and Model 3 (see Figure 1h). It is noted that the Model 1 surface is relatively uneven, with a large difference in the height of the surface profile of the peaks and valleys. Especially in the surface with negative curvature morphology near the right edge. The distribution of peaks and valleys on the surface of Model 2 is relatively uniform. The height difference of the surface profile is relatively stable without a large curvature profile. In comparison to Models 1 and 2, the Model 3 has a larger distance between the peaks and valleys, namely, a larger wavelength. Meanwhile, a larger profile height along with a reduced number of peaks and valleys as well as a nonuniform distribution. In summary, different surface topography characteristics will not only affect the absorption of laser heat, but also have an impact on the molten pool flow and the distribution of driving force and further determine the polished surface quality.

### 2.3. Boundary Conditions

(1)Heat transfer boundary condition

Equation (7) describes the laser beam radiation which induces the thermal convection and surface to surface ambient radiation heat loss whiles the thermal insulation in boundary 4 is expressed in Equation (8) [22,23,24,25,26].
(7)$$-k\nabla T=h\left(T-T_{a}\right)+\varepsilon \sigma \left(T^{4}-T_{a}^{4}\right)$$
(8)∇T=0
where *ε* is the surface emissivity, *σ* is the Stefan-Boltzmann constant, *T_a_* is the ambient temperature, *h* is the convective coefficient.

(2)Momentum boundary condition

Equation (9) describes the velocity field at the boundary which is set as no slip wall in boundary 4 whiles the flow velocity in the r direction which is limited to zero in boundaries 2 and 3 is expressed in Equation (10) [25].
(9)$$u_{r}=u_{z}=0$$
(10)$$u_{r}=0,\;\frac{\partial u_{z}}{\partial r}=0$$
where *u_r_* and *u_z_* is the fluid flow velocity along the *r* and *z* directions.

(3)Free surface boundary condition

Equation (11) describes the total stress act on the free surface (boundary 1) of the molten pool. Additionally, the normal and tangential components are expressed in Equations (12) and (13) [22,23,24,25,26].
(11)$$\sigma =\left(\nabla \cdot \vec{n}\right)\gamma \vec{n}-\nabla \gamma$$
(12)$$\sigma_{n}=\left(\nabla \cdot \vec{n}\right)\gamma \vec{n}=\kappa \gamma \vec{n}$$
(13)$$\sigma_{t}=\nabla \gamma =\frac{\partial \gamma }{\partial T}\nabla_{s}T\cdot \vec{t}=\frac{\partial \gamma }{\partial T}\left[\nabla T-\vec{n}\left(\vec{n}\cdot \nabla T\right)\right]$$
where *γ* is the surface tension coefficient, *κ* is the surface curvature. *∂γ*/*∂T* is the temperature gradient of surface tension, $\vec{n}$ and $\vec{t}$ are represent the normal and tangential vectors, respectively.

### 2.4. Laser Moving Heat Source

The model uses a top-hat laser heat source with uniform energy distribution. Accordingly, the stationary and moving heat source are, respectively, expressed by Equations (14)–(17) [32].
(14)$$Q_{s}=\frac{P}{\pi R_{0}^{2}}$$
(15)$$f(r^{*})=\{\begin{array}{cc} 1 & \left|r^{*}\right|\leq r_{0}\\ 0 & \left|r^{*}\right|\geq r_{0} \end{array}$$
(16)$$r^{*}=r-vt-0.05$$
(17)$$Q_{m}=Q_{s}\times f(r^{*})$$
where *P* is the laser power, *r*_0_ is the spot radius, *r* is the independent variable in the cylindrical coordinate system, *v* and *t* are the laser moving velocity and time, 0.05 (unit: mm) is the starting position of the laser polished surface. Furthermore, the product of piecewise function *f*(*r**) and *Q_s_* describe the laser energy density acting within the laser beam. In addition, the modeling process parameters and specific boundary conditions set in physical fields are shown in Table 2 and Table 3, respectively.

### 2.5. Moving Mesh

The model tracks the deformation of the liquid/gas interface by means of moving mesh. Moreover, coupling of Arbitrary Lagrangian-Eulerian (ALE) method with momentum conservation equation expression is shown in Equation (18) [33].
(18)$$u_{m}\cdot \vec{n}=u_{mat}\cdot \vec{n}$$
where *u_m_* is the moving velocity of the mesh, *u_mat_* is the material velocity.

### 2.6. Mesh and Configurations

In order to calculate the displacement of the free surface accurately, the free surface is hydrodynamically calibrated with a maximum cell size of 0.8 μm and a maximum cell growth rate set to 1.05. Owing to most of the solution area is a solid, a general physical calibration is used in the solution domain with a maximum cell size of 20 μm so as to save computational cost. The maximum cell size is 20 μm, and the maximum cell growth rate is set to 1.1. Most importantly, the moving mesh needs to be Laplace smoothed. If hyper-elasticity or Yeoh smoothing is used, it may lead to singularity and nonconvergence of the top surface mesh [34,35]. The specific mesh parameter for the solution domain as shown in Table 4. The mesh division results are shown in Figure 2. Owing to the different initial morphologies, the number of free triangles in the solution domain varies. The number of mesh cells for models 1, 2 and 3 are 56,496, 56,056 and 57,182, respectively. The average cell quality is above 88% for each model. Meanwhile, the calculation for each model requires about 12 h on a computer equipped with 16.0 GB RAM and four Intel(R) Core (TM) i7-9700K processors at 3.60 GHz CPU speed.

## 3. Experimental Setup and Methods

### 3.1. Polishing Experimental Setup

The purpose of the experimental study is to verify the established numerical model of the LP melt pool dynamics of Ti6Al4V alloy. The schematic and physical diagrams of the polishing experimental system, as shown in Figure 3. The system consists of laser transmitter, beam expander, dynamic focusing system, 2-axis CNC rotary table and protective gas device [36]. The laser transmitter is a single-mode continuous wave fiber laser (Model: MFSC-1000 W, from Shenzhen Chuangxin Laser Co., Ltd., Shenzhen, China) with adjustable power from 150 W to 1000 W [37]. Furthermore, the laser beam passes through the beam expander (Model: 2-8-355-200 m, from Nanjing Wavelength Opto Electronic Pte., Ltd., Nanjing, China) with a top-hat energy distribution, corresponding to a maximum beam quality M^2^ of less than 1.3 [37]. The dynamic focusing system (Model: SDL-F20PRO-3, from Suzhou FEELTEK Laser Technology Co., Ltd., Suzhou, China) with a maximum polishing area of 600 × 600 mm^2^ can obtain a laser beam with a maximum scanning speed of 4000 mm/s, with the focal point of the laser beam generated at 720 mm from the polished surface [36]. Additionally, argon gas, with a purity of 99.99%, is used as a shielding gas to fill the processing chamber to prevent surface oxidation during the polishing process.

### 3.2. Experimental Methods

The material used for the polishing experiments was Ti6Al4V alloy, whose main chemical element compositions are shown in Table 5 [22]. Prior to polishing experiment, the initial surface is cleaned with anhydrous ethanol to prevent dust and other impurities from reducing the polishing quality. Furthermore, in order to ensure that the simulation model is consistent with the experiment, the polishing was performed at the laser focus when the power of 150 W, the scanning speed is 300 mm/s. Finally, the surface morphology of the single-line polished tracks for Models 1, 2 and 3 (see Figure 12a,c,e) was obtained by the 3D optical profilometer (Model: ContourGT-X, from Bruker Nano, Inc., Tucson, AR, USA) based on white light interferometer technology.

## 4. Results and Discussion

### 4.1. Molten Flow Behavior of Model 1

Figure 4 demonstrates evolution of molten pool morphology and the distribution of temperature and velocity fields for Model 1. The polishing was performed at the laser focus when the power of 150 W, the scanning speed is 300 mm/s. The black line is the isotherm of the radiation region, the upper isotherm is the liquidus temperature, the lower is the solidus temperature. The region between the two isotherms is defined as mushy zone [38,39,40]. From the simulation results, it can be seen that the polished surface starts to melt and forms a shallow molten pool of 9 μm on the surface at about 0.3 ms. After which the depth of the molten pool expands continuously with the increase of laser radiation duration until it reaches about 60 μm at 3.5 ms. Furthermore, it is worth noting that the overall molten pool width is also gradually increasing, but at 2.5 ms, it can be found that the extent of the paste zone is smaller compared to 0.5~0.2 ms. It is due to the high cooling rate of the molten pool resulting in a larger area of molten pool solidification at the tail of the spot than the expansion area of the melt front. Moreover, the depth and width of the molten pool continued to increase gradually in the heating duration range from 2.5 ms to 3.5 ms. It reaches a maximum of 58 μm and 435 μm at 3.5 ms. When the laser radiation shut off, the depth and width of the molten pool rapidly decreased to 55 μm and 330 μm after a cooling time of 0.05 ms. The molten pool was completely solidified until 3.8 ms.

### 4.2. Analysis of Temperature Field

From the temperature field distribution, it can be seen that the highest temperature of the polished surface always appears the position near the trailing edge of the laser beam. When the surface temperature above the melting temperature at 0.3 ms, the spot position moves from left to right with the heating duration. Owing to the heat conduction, the surface accumulates much heat, the temperature of the area radiated by the spot gradually increases at 0.3~1.5 ms. Then it reaches the maximum value of about 2760 K. In further, as the laser radiation duration from 1.5 ms to 3.5 ms, surface temperature decreases and is accompanied by small fluctuations. It is due to the expansion of the molten pool area and the increase of the melt flow rate generating larger heat convection and radiation. It results in losing some of the heat, thereby affecting the surface temperature distribution. Simultaneously, it can be observed that the heat affected area of the polished region expands with increasing radiation duration. During the cooling period, the molten pool can no longer obtain heat flux from the laser radiation. With the transfer of heat from the high temperature region of the molten pool to the low temperature region, the surface temperature plummets below the solidus temperature within 0.3 ms after the laser heating is stopped.

### 4.3. Analysis of Velocity Field

Furthermore, it can be seen from the velocity field distribution that the maximum flow velocity always occurs near the solid and liquid phase regions on both sides of the pool. At 0.3 ms, the temperature of the surface peaks reaches above the melting temperature (1900 K) and begins to melt. At this time, the material dynamic viscosity decreases, the capillary force and thermocapillary overcomes the viscous stress and drives the melt to flow from the peaks to the valleys (see Figure 5). It is worth noting that the thermocapillary forces dominate the molten pool in the region of 12 < *r* < 93 μm as tangential flow along the surface. Correspondingly, in the region of 93 < *r* < 150 μm, the capillary forces dominate the melt with large curvature normal to the surface along the peaks towards the valleys. In addition, due to the negative surface tension temperature coefficient of Ti6Al4V alloy, the higher the temperature of the polished surface, the lower the surface tension in the area. Thus, the thermocapillary forces cause the melt to flow from the center of the beam (low surface tension) to the edge of the molten pool (high surface tension). Meanwhile, the temperature gradient near the edge of the trailing molten pool is greater than that at the center of the beam and at the melt front. Therefore, the maximum velocity of the molten pool always occurs at the edge of the molten pool as well as the maximum velocity of the molten pool is 0.35 m/s. At the beginning of LP, namely in the heating range of 0.3 ms to 1.5 ms, the temperature gradient at the trailing edge of the molten pool increases as the surface temperature rises. It results in more pronounced Marangoni convection or thermocapillary forces. Accordingly, the molten pool velocity increases from 0.35 m/s to 1.97 m/s. From 1.5 ms to 3.5 ms, the molten pool flow velocity does not vary much, reaching a maximum flow velocity of about 2.25 m/s at the 3 ms moment. When the laser radiation stops, the temperature gradient in the molten pool area decreases due to the sharp drop in surface temperature. Moreover, the dynamic viscosity of the liquid metal gradually transitions to high viscosity, namely, the viscosity within the molten pool is enhanced. It causes the melt flow rate to drop to 0.44 m/s until the molten pool is completely solidified at 3.8 ms. At this point, the very high dynamic viscosity limits the molten pool velocity to close to zero.

### 4.4. Analysis of Free Surface Evolution

The evolution of the free surface morphology of the molten pool indicates that the shallow part of the surface starts to melt at a polishing duration of 0.3 ms. Due to the small melting area, the molten pool morphology did not change significantly. At 0.5 ms, the trailing melt of the molten pool flows from right to left by Marangoni convection and form half peaks. As the laser beam continues to move to the right, the peaks begin to melt and the melt flows from the peaks to the valleys under the combined effect of normal capillary force, tangential thermocapillary force and body force. At 3 ms, it can be found that the height of the surface profile of the polished peaks is significantly reduced compared to the initial moment and the number of peaks is reduced. It is worth noting that at a polishing time of 3.5 ms, the molten pool surface forms depressions larger than the spot diameter at the edges. It is due to the Marangoni convection formed by the temperature gradient between the center and the edge of the beam causing the melt to flow from the center to the outside. Additionally, the high cooling rate of the molten pool prevented the capillary force from having time to smooth the depressions normally, and a valley with a depression depth of 26 μm was formed on the surface at 3.8 ms.

### 4.5. Analysis of Secondary Surface Bumps Formed

In addition, it can be found that the contour height of the surface peaks has been reduced, but it has not yet reached a smooth effect. In order to analyze the causes of this phenomenon, the evolution duration of the molten pool morphology was further refined. Figure 6 represents the evolution of the molten pool morphology for a polishing time of 0.7~1 ms for Model 1. At 0.7~0.8 ms, a double swirl Marangoni convection can be found on both sides of the molten pool. At the same time, the center of the molten pool is swollen by the buoyancy force, causing a local bump on the surface. At 0.91~0.92 ms, it can be observed that the left side of the molten pool is dominated by capillary forces that make the molten projections flow along the surface normal to the melt. Thus, it results in the melt to be extruded on both sides. Further, a Marangoni convection acting tangentially along the surface was formed at 0.95 ms in the trailing molten pool again. Simultaneously, due to the surface temperature approaching the solid phase temperature, the melt dynamic viscosity increases as well as the capillary force is no longer able to normalize the bulge profile. Thus, the surface is formed a bulge profile at 1 ms. Furthermore, to eliminate or reduce the surface roughness caused by peaks, the laser power can be increased, or the scanning speed can be reduced to increase the survival life of the molten pool. It allows for adequate smoothing of the polished surface by capillary and thermocapillary forces.

### 4.6. Molten Flow Behavior of Models 2 and 3

The evolution of local molten pool morphology and the distribution of temperature field and velocity field for Models 2 and 3 are shown in Figure 7 and Figure 8, respectively. Since this model uses a fixed absorption rate, Models 2 and 3 have similar general patterns in the temperature field compared to Model 1. Namely, the temperature of the polishing area is increased with the increase of radiation time, the highest temperature are in the trailing edge of the beam and stable at about 2700 K. The results show that different initial surface morphologies have a little effect on the temperature field. In addition, for the same input laser energy density, the molten pool temperature field distribution depends mainly on the absorption rate of the laser by the material surface. In terms of velocity field, the flow direction of the molten pool is from the center of the spot to the edge, and the maximum velocity occurs near the melt front.

### 4.7. Evolution of Melt Hydrodynamics for Models 1, 2 and 3

Figure 9 illustrates the variation of the maximum flow velocity of liquid metal in the molten pool region with laser heating and cooling duration during the polishing process. With different initial surface characteristics, the polished surface starts to partially melt at a moment of about 0.3 ms. Thereafter, at 0.3~0.5 ms, the molten pool velocity gradually increases with heating time. It can be noticed that Model 1 has the smallest velocity (2.11 m/s) compared to Model 2 and Model 3 at 0.5 ms. The maximum velocity in the melting region (3.13 m/s) was then reached at 1 ms. Model 2 has the second highest velocity (2.76 m/s) at 0.5 ms compared to Models 1 and 3, followed by a maximum melt velocity (3.15 m/s) at 3.5 ms. Model 3 has the maximum velocity (3.13 m/s) and the maximum melt flow velocity (3.15 m/s) at 1 ms. In the cooling phase, the high cooling rate of the molten pool temperature is accompanied by a sharp increase in the melt dynamic viscosity, which results in a cliff-like decrease in the melt flow rate. Models 1 and 3 have a high dynamic viscosity close to 0 at 3.59 ms. They have a molten pool survival lifetime of about 0.09 ms. Model 2 has a flow rate close to 0 at 3.62 ms and a molten pool survival lifetime of about 0.12 ms. The results show that the overall molten pool velocity under different initial profiles is inverted “U” profile and quasi-stable when the melt reaches a certain velocity. In addition, different surface features influence the moment and location of the maximum velocity of the molten pool. It also has an effect on the survival life of the molten pool, which further affects the final polished morphology.

The variation of the molten pool depth and width in the polished area with the laser radiation duration for different initial morphological features, as shown in Figure 10. It can be found that the molten pool depths of Model 2 at radiation times of 0.5~1.4 ms are slightly greater than those of Models 1 and 3. At 2~3 ms, the melting depths of Models 1 and 3 showed a decreasing trend. In contrast, the Model 2 melt depth increases with time but more slowly. Finally, Models 1 and 3 start to rise at 3.5 ms and the melting depth increases to 58 μm and 47 μm, respectively. But the Model 2 melting depth slightly decreased to 49 μm. The results show an overall increasing trend of molten pool depth with time during the polishing process. However, the different surface morphological features affect the mass flow of the melt in the z-direction, thus, affecting the heat transfer inside the molten pool. It further causes the melt depth to fluctuate with the laser radiation time. Additionally, the general pattern of the molten pool width parallel to the polishing direction with time expressed in Figure 10b is similar to that of the melt depth. Furthermore, as the cooling rate of the molten pool tail temperature is greater than the rate of expansion displacement of the melt front. It can be found that the melt widths of Models 1, 2 and 3 all drop locally during the polishing process.

Figure 11 demonstrates the profile height of the polished surface in different initial morphologies. It can be seen from the figure that the height of the half-wave profile at the starting position of the surface after LP is the largest (24 μm) for Model 1, the second (13.5 μm) for Model 3 and the smallest (1.6 μm) for Model 2. It maps the melt to the strongest Marangoni convection at this location. In addition, it can be found that the surface curvature at the starting position of the unpolished morphology of Model 2 is the largest (see Figure 1). It illustrates the normal smoothing effect of the capillary force is the most obvious, which basically eliminates the surface bulge. It can be further observed that the polished surface is not eliminated by the normal and tangential thermocapillary forces despite the reduction in the number of peaks and the height of the profile. At the same time, the local surface height is higher than the initial surface, which is due to the short time that the molten pool is in the molten state and the capillary force is too late to normalize the large curvature surface. Meanwhile, the Marangoni convection at the tail of the molten pool causes the melt to flow along the center of the beam towards the edge. As a result, the peaks are not effectively smoothed, and the surface formed local bulge after polishing. In addition, the different surface curvature due to the difference in initial morphology affects the dominant role of capillary forces on the molten pool flow as well as the dominant role of thermocapillary forces after the molten pool is fully developed.

Finally, it can be noted that due to the Marangoni convection of the double spiral nest, the polished end position forms a depression larger than the diameter of the beam. The depression width and depth of Model 2 are the smallest, about 312 μm and 12 μm. Model 3 is the second, about 380 μm and 13 μm. In addition, Model 1 is the largest, about 520 μm and 26 μm. It can also be seen from the above data that the velocity and direction of the flow to the valleys during peaks melting is relatively stable due to the relatively uniform distribution of the surface morphology at the initial surface end of Model 2. The fluctuations generated on the surface of the molten pool are smaller, and a better effect of melt peak filling can be achieved. Thus, the formed depressions are uniform in profile and small in extent. In contrast, the peaks distribution at the end position of the initial surface of Model 1 has an overall negative curvature distribution. This is coupled with the effect of Marangoni convection tangential along the surface. It results in a large depression depth and width. The above results indicate that the initial surface features with the uniform distribution can enhance the smoothing effect.

### 4.8. Experimental Validation

The surface morphology of the single-line polished tracks and comparison between simulated surface height profile and experimental laser polished profiles for Models 1, 2 and 3, as shown in Figure 12. Fourier filter is performed on the 3D morphology of the polished surface so that ensuring the accuracy of the experimental surface profiles. The processing method is the same as described in Section 2.2. The red lines in the wire frame are the single-line polished tracks morphology for Models 1, 2 and 3. The yellow lines indicate the surface line profiles along the center of polished track area (see Figure 12a,c,e). Correspondingly, the red lines indicate the polished surface profiles predicted by numerical simulation, the blue lines represent polished surface profiles by experiments as well as the black represent the initial surface morphology. From the overall profiles of the polished surfaces, the surface asperities were not completely eliminated by means of CW fiber laser polishing. Simultaneously, the positive and the negative curvature profiles were formed nearby the starting position and at the edges of the polished surface. Additionally, it can be observed that the experimental depth of the depressions for Models 1, 2 and 3 are about 24 μm, 14 μm and 15 μm at 1 mm of the polished surface. Correspondingly, the numerical prediction with the depth of about 26 μm, 12 μm and 13 μm. The results indicate that the models surface profile agree reasonably well with the experimental data.

## 5. Conclusions

In this work, we developed a transient 2D model coupled of heat transfer and fluid fields based on moving heat source. It is illustrated that the effect of different initial morphologies on the evolution of the melt hydrodynamics during LP. The following conclusions can be made:(1)The model demonstrated that the complex evolution of the melt hydrodynamics involving heat conduction, thermal convection, thermal radiation, melting and solidification during laser polishing.(2)The uniformity of the initial surface peaks and valleys distribution is positively correlated with the smoothing quality of the polished surface, but has less effect on temperature field, velocity field, as well as melt depth and width of the molten pool.(3)The surface rough profiles are not completely eliminated by capillary and thermocapillary forces due to the high cooling rate of the molten pool, resulting in the formation of secondary surface roughness. It was revealed that the short lifetime of the molten pool is the main reason why the surface bumps are not completely eliminated.(4)The numerical prediction of the depressions for Models 1, 2 and 3 are approximate 26 μm, 12 μm and 13 μm at about 1 mm on the polished surface. Accordingly, the experimental molten pool depths are about 24 μm, 14 μm and 15 μm as well as the errors are approximately 8.3%, 14.3% and 13.3%, respectively.(5)The model not only predicts the morphological evolution of different surfaces from rough to smooth in laser polishing, but also can be suitable for guiding the optimization of polishing parameters such as laser power and scanning speed. Additionally, this model can be applied to most metallic materials in laser polishing.

## Figures and Tables

**Figure 1 micromachines-12-00581-f001:**
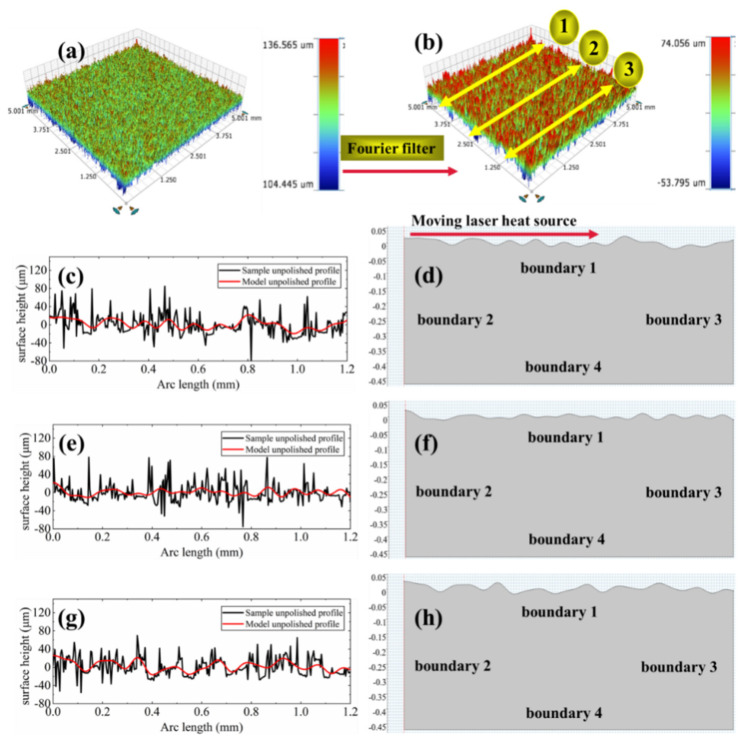
Geometry model of molten pool. (**a**) Optical morphology of initial surface; (**b**) Optical morphology of initial surface after Fourier filtering; (**c**) Surface profile height of Model 1; (**d**) Geometry of Model 1; (**e**) Surface profile height of Model 2; (**f**) Geometry of Model 2; (**g**) Surface profile height of Model 3; (**h**) Geometry of Model 3.

**Figure 2 micromachines-12-00581-f002:**
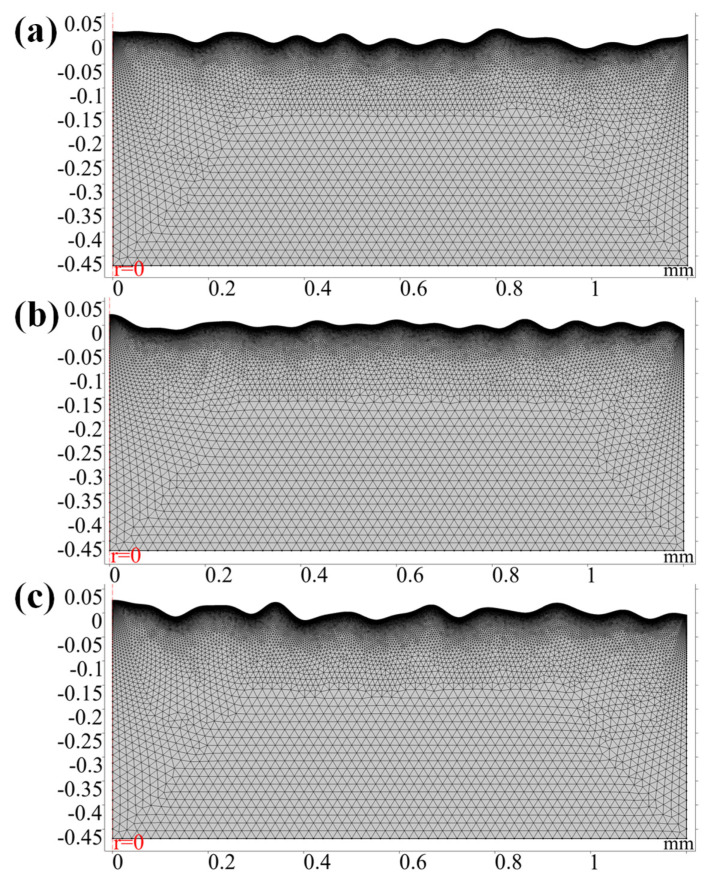
Meshed geometry. (**a**) Model 1; (**b**) Model 2; (**c**) Model 3.

**Figure 3 micromachines-12-00581-f003:**
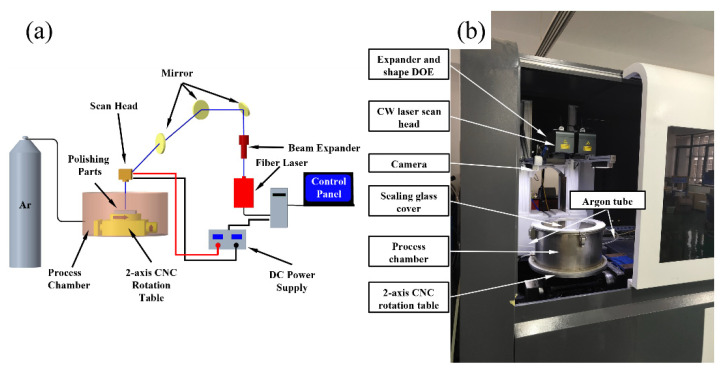
Experimental setup and device of the polishing on Ti6Al4V. (**a**) Experiment device; (**b**) Principle of experiment device.

**Figure 4 micromachines-12-00581-f004:**
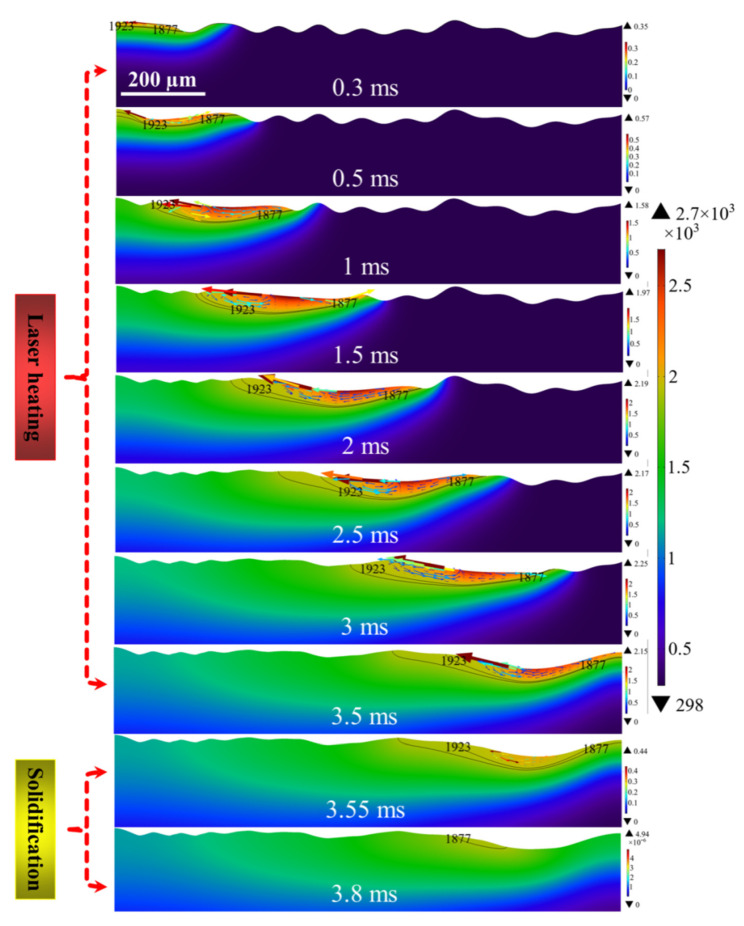
Evolution of molten pool morphology and the distribution of temperature field (color surface contour, unit: K) and velocity field (colored arrow plots, unit: m/s) of Model 1.

**Figure 5 micromachines-12-00581-f005:**
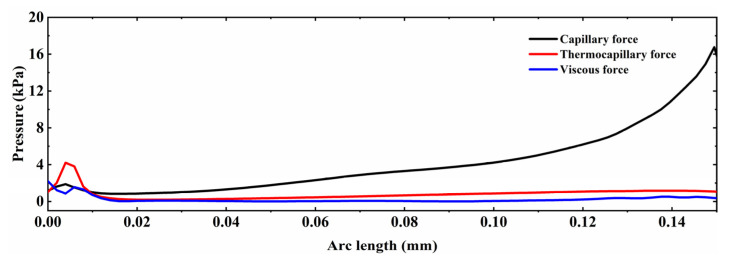
The dominant of capillary and thermocapillary forces as well as viscosity distribution at 0.3 ms heating duration.

**Figure 6 micromachines-12-00581-f006:**
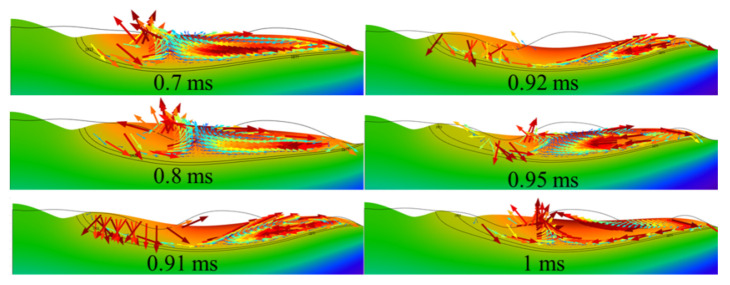
Evolution of local molten pool morphology and the distribution of temperature field (color surface contour, unit: K) and velocity field (colored arrow plots, unit: m/s) of Model 1.

**Figure 7 micromachines-12-00581-f007:**
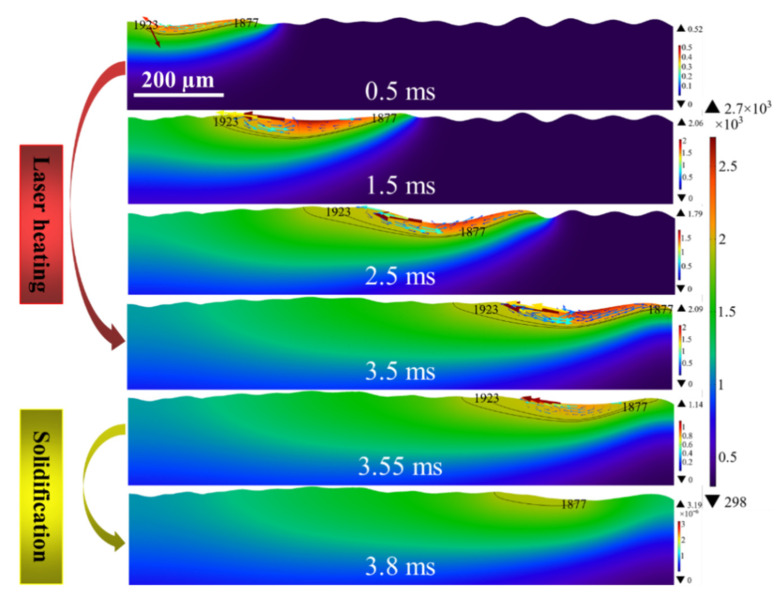
Evolution of molten pool morphology and the distribution of temperature field (color surface contour, unit: K) and velocity field (colored arrow plots, unit: m/s) of Model 2.

**Figure 8 micromachines-12-00581-f008:**
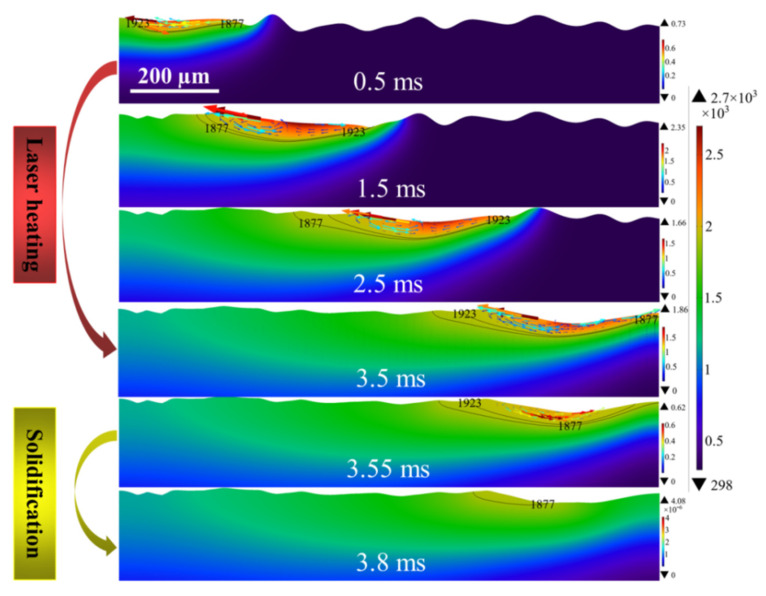
Evolution of molten pool morphology and the distribution of temperature field (color surface contour, unit: K) and velocity field (colored arrow plots, unit: m/s) of Model 3.

**Figure 9 micromachines-12-00581-f009:**
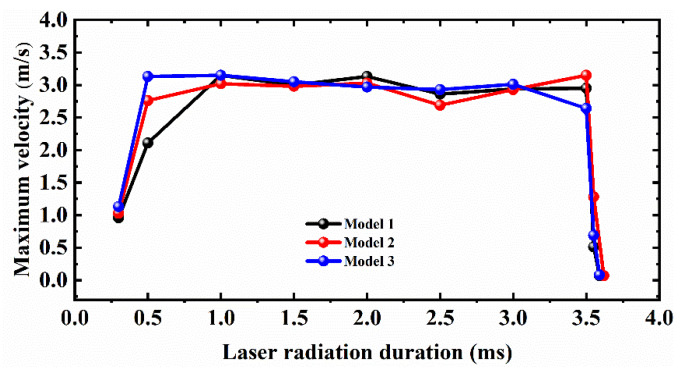
Variation of the maximum velocity of liquid metal in the polished area with radiation duration.

**Figure 10 micromachines-12-00581-f010:**
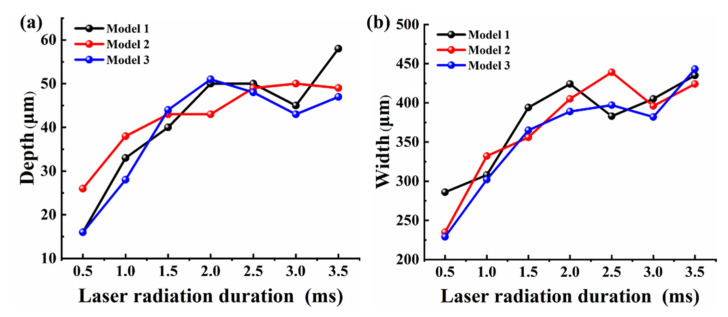
Variation of molten pool depth and width in the polished area with radiation duration: (**a**) melting depth; (**b**) melting width.

**Figure 11 micromachines-12-00581-f011:**
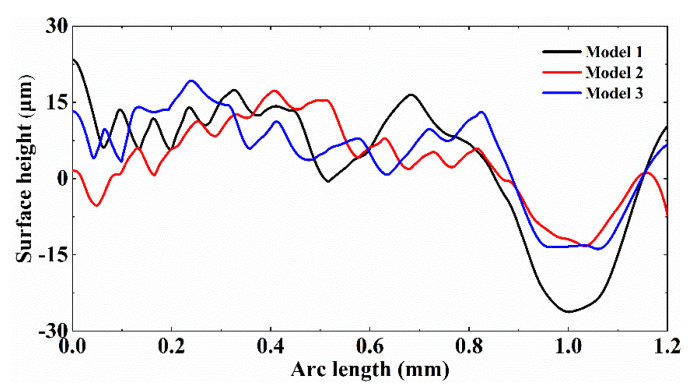
Profile height of the polished surface.

**Figure 12 micromachines-12-00581-f012:**
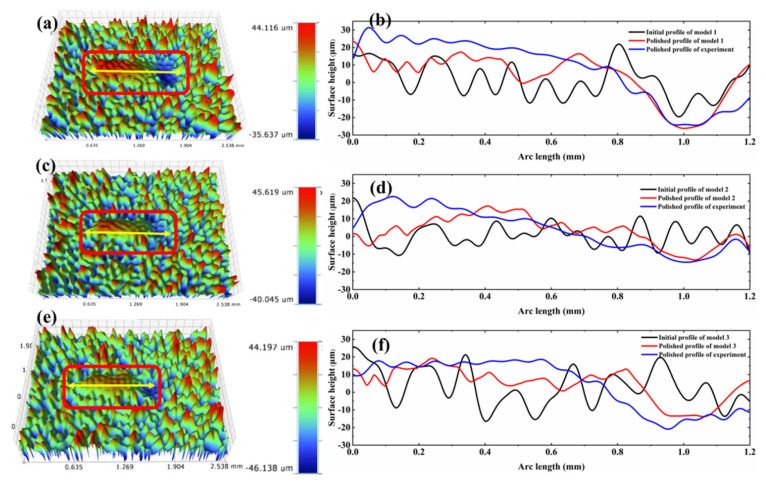
The surface morphology of the single-line polished tracks and comparison between simulated surface height profile and experimental laser polished profiles for Models 1, 2 and 3. (**a**) Polished optical morphology of Model 1; (**b**) Surface profile height of Model 1; (**c**) Polished optical morphology of Model 2; (**d**) Surface profile height of Model 2; (**e**) Polished optical morphology of Model 3; (**f**) Surface profile height of Model 3.

**Table 1 micromachines-12-00581-t001:** Thermophysical properties of Ti6Al4V [22,23,24,25,26,27].

Parameter	Nomenclature	Value
Solidus temperature (K)	*T_s_*	1877
Liquidus temperature (K)	*T_l_*	1923
Melting temperature (K)	*T_m_*	1900
Boiling temperature (K)	*T_b_*	3315
Ambient temperature (K)	*T_a_*	298.15
Solidus density (kg m^−3^)	*ρ_s_*	4420
Liquidus density (kg m^−3^)	*ρ_l_*	4000
Dynamic viscosity (Pa s)	*μ*	0.005
Solidus thermal conductivity (Wm^−1^ K^−1^)	*k_s_*	21
Liquidus thermal conductivity (Wm^−1^ K^−1^)	*k_l_*	29
Solidus specific heat (Jkg^−1^ K^−1^)	*C_p-s_*	670
Liquidus specific heat (Jkg^−1^ K^−1^)	*C_p-l_*	831
Convective coefficient (Wm^−2^ K^−1^)	*h*	10
Temperature derivative of surface tension (Nm^−1^ K^−1^)	*∂γ/∂T*	−2.8 × 10^−4^
Latent heat of melting (Jkg^−1^)	*L_m_*	2.86 × 10^5^
Emissivity	*ε*	0.6
Absorptivity	*α* _0_	0.3

**Table 2 micromachines-12-00581-t002:** Modeling process parameters in LP.

Polishing Parameter (Unit)	Nomenclature	Value
Laser beam radius (mm)	*R* _0_	0.135
Laser power (W)	*P*	150
Laser scanning speed (mm s^−1^)	*v*	300
Laser heating duration (ms)	*t_h_*	3.5
Cooling duration (ms)	*t_c_*	0.3

**Table 3 micromachines-12-00581-t003:** Summary of physical field boundary conditions.

Boundary Condition	Boundary (See Figure 1)	Physical Condition
Boundary heat source	1	Laser radiation
Convection	1, 2, 3	Natural convection
Diffuse surface	1, 2, 3	Radiation
Thermal insulation	4	Insulation
Capillary force	1	Weak contribution
Themocapillary force	1	Marangoni effect
Wall	2, 3,4	No slip wall

**Table 4 micromachines-12-00581-t004:** Free triangular element size parameters.

Parameter (Unit)	Top Layer	The Rest
Maximum element size (μm)	0.8	20
Minimum element size (μm)	0.002	0.024
Maximum element growth rate	1.05	1.1
Curvature factor	0.2	0.2

**Table 5 micromachines-12-00581-t005:** Main chemical composition of Ti6Al4V (wt %).

Ti	Al	V	C	Fe	O	N
Balance	5.50–6.75	3.50–4.50	0.08	0.30	0.20	0.05

## Data Availability

Not applicable.

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
