# Peer review of "Numerical Simulation of Effect of Different Initial Morphologies on Melt Hydrodynamics in Laser Polishing of Ti6Al4V"

_micromachines, 2021, doi:10.3390/mi12050581_

Round 1

Reviewer 1 Report

I ask you humbly to check attached file

Author Response

We were pleased to have an opportunity to revise our paper entitled, “Numerical simulation of effect of different initial morphologies on melt hydrodynamics in laser polishing of Ti6Al4V” Ref. No.: micromachines-1230668. In revising the paper, we have carefully considered your comments and suggestions, as well as those of the reviewers. As instructed, we have attempted to succinctly explain changes made in reaction to all comments and have replied to each comment in point-by-point fashion. The revised contents have been shown with a yellow background in this attached manuscript. The corresponding changes and refinements made in the revised paper are shown in the attachment.

Reviewer 2 Report

The reviewer comments of the paper «Numerical simulation of effect of different initial morphologies on melt hydrodynamics in laser polishing of Ti6Al4V»- Reviewer

The authors presented an article «Numerical simulation of effect of different initial morphologies on melt hydrodynamics in laser polishing of Ti6Al4V». The article is interesting and well written. However, there are several points in the article that require further explanation.

Comment 1:

The abstract needs to be completed. Demonstrate in the abstract novelty, practical significance. Add quantitative and qualitative work results to the abstract. Add material Ti6Al4V in the abstract.

Highlights are superfluous for MDPI.

Comment 2:

In general, the introduction is written clearly and clearly.

However, it will be helpful to add a few recent articles published in the last 3 years.

Comment 3:

  1. Numerical simulation

Are all formulas original? If not, please provide relevant citations.

Show on the figure the boundary conditions.

What type of finite elements have you chosen and why? Justify this.

Comment 4:

  1. Experimental setup and methods

Are all figures original? If not needed appropriate citations and permissions.

For measurement devices, software and machines used in research, indicate in parentheses (manufacturer, city, country).

Comment 5:

It will be useful to add a section of Nomenclature in which to sign all the physical quantities and abbreviations encountered in the article. There are many physical quantities in the text and such a section will help to find the description of the necessary element.

For example,

  • : Density (g/cm3)

LP             : Laser polishing

etc.

Comment 6:

The conclusions need to be improved.

What is the novelty of the article? What is the practical significance? What are the differences from previous works?

Provide quantitative and qualitative conclusions for each parameter under study.

Conclusions should reflect the purpose of the article.

The article is interesting. However, the article needs to be improved. Authors should carefully study the comments and make improvements to the article step by step. All changes should be highlighted in color. After major changes can an article be considered for publication in the "Micromachines".

Author Response

(The authors gave the same response as above.)

Round 2

Reviewer 1 Report

Paper can be accepted

Author Response

Thanks for your comments again.

Reviewer 2 Report

The authors have improved the article according to the comments.
However, before accepting, it is important to address several issues:
1. Group citations [1-6] are best broken down into several sentences.
It is useful to add articles:
International Journal of Advanced Manufacturing Technology 2017,
89(9-12), 3149-3159. DOI: 10.1007/s00170-016-9216-x
Surface and Coatings Technology 2019, 380, 125016. DOI: 10.1016/j.surfcoat.2019.125016
Journal of Materials Research and Technology 2021, 11, 719-753. DOI:
10.1016/j.jmrt.2021.01.031
Journal of Manufacturing Processes 2021, 65, 478-490. DOI: 10.1016/j.jmapro.2021.03.045

2. Fill out the list of references in accordance with the MDPI requirements.
After elimination of these comments, the article can be accepted for
publication.

Author Response

We were pleased to have an opportunity to revise our paper entitled, “Numerical simulation of effect of different initial morphologies on melt hydrodynamics in laser polishing of Ti6Al4V” Ref. No.: micromachines-1230668. In revising the paper, we have carefully considered your comments and suggestions, as well as those of the reviewers. As instructed, we have attempted to succinctly explain changes made in reaction to all comments and have replied to each comment in point-by-point fashion. The revised contents have been shown with a yellow background in this attached manuscript. Please see the attachment.
